🔓 | **Open Peer Review** | Clinical Microbiology | New-Data Letter

# Building of a new Spectra for the identification of *Phytobacter* spp., an emerging Enterobacterales, using MALDI Biotyper

A. Dal Lin,[1,2] D. O. Kulek,[1,2] G. A. Gonçalves,[1] L. Kraft,[1] J. F. C. Neto,[1] G. Vizentainer,[2] M. Pillonetto[1,2]

**KEYWORDS** *Phytobacter*, Enterobacteriaceae, mass spectrometry, MALDI-TOF, phenotypic identification, emerging pathogen

*Phytobacter* is an emerging pathogen responsible for different human infections (i.e., sepsis and urinary tract infection), including outbreaks (1). The first reported human infection was an outbreak of contaminated total parenteral nutrition in Brazil in 2013 (2). Recently, it was recognized as an emerging pathogen by references (3, 4) and in one of the most comprehensive textbooks of Clinical Microbiology (5). Most identification methods, such as manual biochemical tests (Enterobacterales kit - Newprov, Brasil; API 20E - bioMérieux, France), different automation panels (Vitek-2 GN - bioMérieux, France, MicroScan (Beckman Coulter, Inc., West Sacramento, CA, USA), and matrix-assisted laser desorption/ionization time-of-flight mass spectrometry (MALDI-TOF-MS, Vitek-MS), fail to identify this genus, assigning these isolates mistakenly to *Pantoea agglomerans*, *Kluyvera intermedia*, *Leclercia adecarboxylata*, *Salmonella* sp., *Klebsiella aerogenes*, *Klebsiella oxytoca*, *Pluralibacter gergoviae*, *Cronobacter sakazakii*, *Cronobacter turicensis*, and *Cronobacter* sp. (1, 2, 6–8). Even sequencing methods, when not properly implemented or the data not correctly analyzed, can misidentify isolates from the genus as *Pseudescherichia vulneris* and "*Citrobacter bitterni*" (BLAST for 16S rDNA) or *K. intermedia* and *Enterobacter* spp. [mistaken identification data using whole-genome sequencing—WGS—deposited at GenBank; (1, 9) Pillonetto M, personal communication]. Therefore, there is an urgent need to optimize these identification systems. The present study aimed to build a new MALDI-TOF-MS library to identify *Phytobacter* spp. correctly.

We used MALDI Biotyper Sirius (Bruker Daltonics, Bremen, Germany) to construct a new Spectra. Four strains of *Phytobacter ursingii*, 12 *Phytobacter diazotrophicus*, and 3 isolates of a novel, undescribed *Phytobacter* species (not yet published, manuscript in preparation) were used, including *P. diazotrophicus* ATCC 27990 and *P. diazotrophicus* DSM 17806[T]. The isolates came from a reference laboratory in Brazil (LACEN/PR). Initially, suspected strains were gram-negative, strong lactose fermenting, with triple-negative results for lysine, ornithine, and arginine, positive for indole, citrate, and Voges-Proskauer (VP) but negative for the fermentation of inositol and melibiose. All strains had their identification confirmed previously by WGS. The mass spectra were analyzed and compared with other strains in the reference database. Tests were carried out in octuplicate using 19 strains among the three species. The indirect method was used to obtain better quality spectra, where an extraction with acetonitrile was performed before submitting the strains to MALDI, according to the methodology recommended by the manufacturer. The software used for data analysis was Bruker Compass Data Analysis 2.0, version 3.5.

Differences between species were observed. Figure 1 shows that *P. diazotrophicus* has spectral peaks from 3,000 to 11,000. For *P. ursingii*, the peaks are more stable and linear, starting from 2,000 to 11,000. For the *Phytobacter* sp. *nov*, mass formation starts from 3,000 and goes up to 16,000. All profiles were unique, as shown in Fig. 1. Table S1a and

Address correspondence to M. Pillonetto, marcelopilonetto@gmail.com.

The authors declare no conflict of interest.

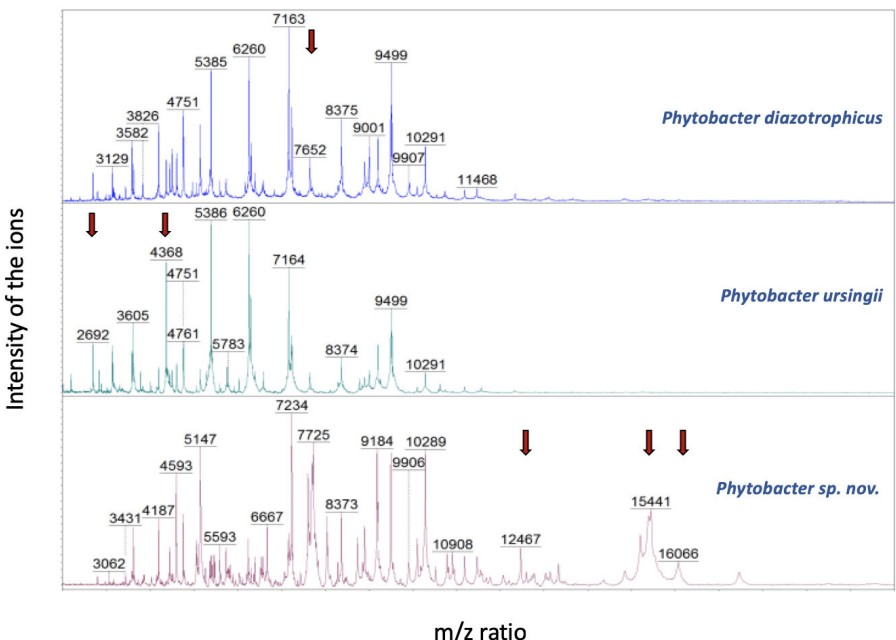

**FIG 1** Multiple spectra report of three different *Phytobacter* species, with some distinctive peaks between them, marked with red arrows. In MALDI-TOF-MS spectra, the x-axis represents m/z ratio, and the y-axis represents the intensity (or number) of same/similar ions.

b show a clear improvement in the correct identification at species level, after the new library was implemented, rising from two to nine correct species identified.

According to our knowledge, Vitek-MS MALDI-TOF (bioMerieux, France) is still missing the public *In Vitro* Diagnostics (IVD) database for *Phytobacter* spp. identification. The exception is an Research Use Only (RUO) database created successfully at our lab (LACEN/PR), where it correctly identified 96% (24/25) of the isolates (Mazzetti A, personal communication). Conversely, a mass profile of *P. ursingii* was recently uploaded to a new database for MALDI Biotyper Sirius. This occurred in April 2023 and subsequently supported the identification of two outbreaks caused by *Phytobacter* spp. in a kidney diseases clinic in Brazil (Kulek DO, 2024—personal communication) and also in a Neonatal Intensive Care Unit (NICU) in Japan (8), as well as a sepsis case in Iowa, USA (10), where *P. ursingii*, identified only by MALDI-TOF-MS, was attributed as etiological agent, but since WGS was not performed, its final ID to species level could not be confirmed. Human isolates of *P. diazotrophicus* have also been reported in Argentina and China (11, 12), showing that although the current Bruker profile cannot differentiate between species, it can alert the user to the suspicion of *Phytobacter* spp. Further molecular tests (16S rDNA or WGS) are needed to confirm its identification at the species level. In Japan's outbreak, it was interesting to observe that a first run in MALDI Biotyper Sirius gave as possible results *C. sakazakii* or *P. gergoviae* with low identification scores (<2.0). But, on a second run, after the equipment database was updated with the MALDI Biotyper (MBT) Compass reference library, MBT-BDAL-10833 (Bruker), the identification was *P. ursingii* with a score ≥1.99 for all four isolates. WGS confirmed that it was a *Phytobacter* spp. belonging to a different species—*P. diazotrophicus* (8). We expect that with the introduction of the present proposed mass profile, it will be possible to differentiate *Phytobacter* species, especially the yet-to-be-validly published "*Phytobacter* sp. nov." as it has very distinctive peaks at m/z 15441 and 16066.

Correct identification of *Phytobacter* spp., which is increasingly reported worldwide, is paramount. Its misidentification as other *Enterobacterales* can hinder outbreak investigations and may lead to incorrect treatments or source identification. Although many

strains still need to be validated for an exact differentiation between species, a close analysis of the individual characteristics of the mass weights among the three species is possible. This is an essential advance for including the species *P. diazotrophicus* and "*Phytobacter* sp. nov." in the Biotyper database since there is only a library entry for *P. ursingii*, which has been improved in this study, showing different peaks from the other species.

## AUTHOR AFFILIATIONS

[1]Laboratório Central do Estado do Paraná, LACEN/PR, Curitiba, Paraná, Brazil
[2]Pontificia Universidade Católica do Paraná, Curitiba, Paraná, Brazil

## AUTHOR ORCIDs

M. Pillonetto ⓘ http://orcid.org/0000-0003-2896-3186

## AUTHOR CONTRIBUTIONS

A. Dal Lin, Formal analysis, Investigation, Methodology, Validation | D. O. Kulek, Conceptualization, Formal analysis, Investigation, Methodology, Writing – original draft, Writing – review and editing | G. A. Gonçalves, Formal analysis, Investigation, Methodology, Visualization, Writing – original draft, Writing – review and editing | L. Kraft, Formal analysis, Investigation, Methodology | J. F. C. Neto, Formal analysis, Methodology, Writing – review and editing | G. Vizentainer, Formal analysis, Writing – original draft, Writing – review and editing | M. Pillonetto, Conceptualization, Formal analysis, Investigation, Methodology, Resources, Supervision, Validation, Writing – original draft, Writing – review and editing

## ADDITIONAL FILES

The following material is available online.

### Supplemental Material

**Supplemental material (Spectrum01107-24-S0001.docx).** Tables S1 and S2.

### Open Peer Review

**PEER REVIEW HISTORY (review-history.pdf).** An accounting of the reviewer comments and feedback.

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
