## [Reviewer comments · Microbiology Spectrum]

Microbiology Spectrum

Building of a New Spectra for the Identification of *Phytobacter* spp, an Emerging Enterobacterales, Using Maldi Biotyper

Amanda Dal Lin, Debora Kulek, Geiziane Goncalves, Leticia Kraft, José Ferreira da Cunha Neto, Geovana Vizentainer, and Marcelo Pilonetto

Corresponding Author(s): Marcelo Pilonetto, Pontificia Universidade Catolica do Parana Escola de Medicina e Ciencias da Vida

Review Timeline:

Submission Date:	May 2, 2024
Editorial Decision:	May 18, 2024
Revision Received:	July 17, 2024
Editorial Decision:	July 23, 2024
Revision Received:	July 30, 2024
Accepted:	August 4, 2024

Editor: Karen Carroll

Reviewer(s): Disclosure of reviewer identity is with reference to reviewer comments included in decision letter(s). The following individuals involved in review of your submission have agreed to reveal their identity: Clement K.M. Tsui (Reviewer #2)

Transaction Report:

DOI: <https://doi.org/10.1128/spectrum.01107-24>

Re: Spectrum01107-24 (Building of a New Spectra® for the Identification of *Phytobacter* spp, an Emerging Enterobacterales, Using Maldi Biotyper®)

Dear Dr. Marcelo Pillonetto:

Thank you for the privilege of reviewing your work, which has now been reviewed by two experts on this topic. The manuscript requires major revisions before it can be considered acceptable for publication. The reviewers have provided excellent instructions on how to improve the manuscript, which mirror my own suggestions. Once the comments have been thoroughly addressed, which may include the addition of more data, I will then move the manuscript forward for publication.

Below and attached you will find the reviewers' comments and instructions from the Spectrum editorial office.

Revision Guidelines

Sincerely,
Karen Carroll
Editor
Microbiology Spectrum

Reviewer #1 (Comments for the Author):

The manuscript describes the building of a New Spectra® needed to identify three species belonging to the genus *Phytobacter* in the Bruker MALDI-ToF system. Given the previous history of misidentifications of isolates of these species, this is an important dataset that must imperatively be included in the Biotyper Database in order to prevent further errors and oversights, especially in clinical diagnostics. The report is generally well written but it still has some room for improvement concerning clarity and data presentation. Suggestions are included directly as comments in the pdf file.

Reviewer #2 (Comments for the Author):

The paper written by Dal Lin et al. reported the building a new MALDI-tof MS library for *Phytobacter* species. However, this short note was not written well. The paragraphs were disorganized, with missing information. Several sections/claims required elaboration and support. (please includes line number for reviewers).

Major Comments/questions:

1. How many isolates were being used to establish the MALDI-tof MS library? Were type species/isolates included? From different countries? How did you validate the ID initially before using WGS? e.g any phenotypic features on agar compared to other Gram -ve
2. Better to describe an example on how the *Phytobacter* isolates were misidentified using 16S and WGS data.
3. The fact that a library is available for *P. ursingii* should be moved to the introduction (now it was mentioned in the last sentence). Have you compared the spectra of your *P. ursingii* strains to the commercial available spectra?
4. MALDI, MALDI tof , MALDI tof MS were used interchangeably. I believe they meant different things, and sometimes they were used incorrectly. The technique should be called MALDI-tof MS.
5. What is the x- and y- axes of the MS spectra (Fig.1)? The report of spectra variation (3rd paragraph) should be revised properly. These are peptides of m/z showing different intensity.
6. is it possible to include the scores of these validation strains in the supplementary materials?
7. Three references were not yet published. They should be removed or cited as (pers comm), and discussed with caution.

Building of a New Spectra[®] for the Identification of *Phytobacter* spp, an Emerging Enterobacterales, Using Maldi Biotyper[®]

Dal Lin A^{1,2} ; Kulek DO^{1,2}; Gonçalves GA¹; Kraft L¹, Neto JFC¹; Vizontainer G²; Pillonetto M.^{1,2}

¹Laboratório Central do Estado do Paraná, LACEN/PR. São José dos Pinhais, Parana, Brazil

²Pontificia Universidade Católica do Paraná. Curitiba, Paraná, Brazil

Phytobacter spp is an emerging pathogen responsible for different human infections (i.e., sepsis and UTI), including outbreaks (Smits T et al., 2022). The first reported human infection was an outbreak of contaminated Total Parenteral Nutrition in Brazil in 2013 (Pillonetto M et al., 2018a). Recently, it was recognized as an emerging pathogen by Janda & Abbott (2021), Munson & Carroll (2021), and one of the most expressive textbooks of Microbiology - the Manual of Clinical Microbiology (2023). Most identification methods, such as manual biochemical tests, different automation panels, and MALDI-TOF, fail to identify this genus, naming these isolates equivocally as *Pantoea agglomerans*, *Kluyvera intermedia*, *Leclercia adecarboxylata*, *Salmonella* sp., *Klebsiella aerogenes*, *Klebsiella oxytoca*, *Pluralibacter gergoviae*, *Cronobacter sakazakii*, *Cronobacter turicensis* and *Cronobacter* sp. (Pillonetto M et al., 2018a; Pillonetto M et al. 2018b, Smits T et al., 2022; Hon P et al. 2023; Kubota H et al., 2023). Even sequencing methods can misidentify it as *Pseudomonas* spp. and “*Citrobacter bitterii*” (16S rDNA) or *K. intermedia* and *Enterobacter* spp. (Whole Genome Sequencing – WGS) (Liu L et al., 2020; Smits T et al., 2022; Pillonetto M, personal communication). Therefore, there is an urgent need to optimize these identification systems. The present study aimed to build a new MALDI library to identify *Phytobacter* spp correctly.

We used MALDI Biotyper[®] Sirius (Bruker Daltonics, Bremen, Germany) to construct a new Spectra[®]. Strains of *P. ursingii*, *P. diazotrophicus*, and *Phytobacter* sp. nov. (not yet published) were used. The isolates came from a reference laboratory in Brazil (LACEN/PR), and all strains had their identification confirmed previously by WGS. The mass spectra were analyzed and compared with other strains in the reference database. Tests were carried out in octuplicate using 19 strains among the three species. The indirect method was used to obtain better quality spectra, where an extraction with acetonitrile was performed before submitting the strains to MALDI, according to the methodology recommended by the fabricants. The software used for data analysis was Bruker Compass[®] DataAnalysis 2.0, version 3.5.

Differences between species were observed. Figure 1 shows that *P. diazotrophicus* has spectral peaks from 3,000 to 11,000. For *P. ursingii*, the peaks are more stable and linear, starting at 2,000 to 11,000. Furthermore, for the *Phytobacter* sp. nov, mass formation starts at 3,000 and goes up to 16,000, giving a unique profile.

According to our knowledge, Vitek-MS[®] MALDI-TOF (bioMérieux, France) is still missing the public IVD database for *Phytobacter* spp identification. The exception is an RUO database created successfully at our lab (LACEN/PR), where it correctly identified 96% (24/25) of the isolates (Mazzetti A et al. 2024 – manuscript in preparation). Conversely, a mass profile of *Phytobacter ursingii* was recently uploaded to a new database for MALDI Biotyper[®] Sirius. This actualization occurred in April 2023 and already supported the identification of two outbreaks caused by *Phytobacter diazotrophicus* in a Kidney Diseases Clinic in Brazil (Gonçalves GA et al., 2023 – submitted) and also in a NICU in Japan (Kubota et al., 2023), as well as a sepsis case in Iowa, USA (Choice S et al. 2024), that still

need to be confirmed, regarding its real species, ~~one~~ WGS was not processed. Although the **current profile** cannot differentiate between species, it already alerts to the suspicion of the genera *Phytobacter*, ~~guaranteeing~~ that further molecular tests (16S rDNA or WGS) are needed to confirm its identification at the species level. In Japan's outbreak, it was interesting to observe that a first run in MALDI Biotyper® Sirius gave as possible results: *Cronobacter sakazakii* or *Pluralibacter gergoviae* with low identification scores (<2.0). But, on a second run, after the equipment database was updated with the MBT Compass reference library, MBT-BDAL-10833 (Bruker), the identification was *Phytobacter ursingii* with a score ≥ 1.99 for all four isolates. WGS confirmed it was a *Phytobacter* spp, but belonging to a different species - *P. diazotrophicus*. We expect that with the introduction of the present proposed mass profile, it will be possible to differentiate *Phytobacter* species, especially the yet-to-be-validatedly published "*Phytobacter brasiliensis* sp. nov." ~~because~~ it has very distinctive peaks at m/z **1544 and 1606**.

Correct identification of *Phytobacter* spp, increasingly reported worldwide, is paramount. Its misidentification with other Enterobacterales can mask outbreak investigation and even induce incorrect treatments or source investigation. Although many strains still need to be tested for an exact differentiation between species, a close analysis of the individual characteristics of the mass weights between the three species is possible. This is an essential advance for including the species *P. diazotrophicus* and "*Phytobacter brasiliensis* sp. nov." in the Biotyper® database since there is only a library for *P. ursingii*, which has also been **improved** in this study.

Figure 1 - Multiple Spectra Report of three different *Phytobacter* species, **showing clear different spectra** between them.

References

Choice S, et al. BMJ Case Rep 2024;17:e258384. doi:10.1136/bcr-2023-258384

Hon P, Ko KKK, Zhong JCW, De PP, Smits THM, Low J, Vasoo S, Tsui CKM. Genomic Identification of Two *Phytobacter diazotrophicus* Isolates from a Neonatal Intensive Care Unit in Singapore. Microbiol Resour Announc. 2023 Jun 20;12(6):e0016723. doi: 10.1128/mra.00167-23.

Janda JM, Abbott SL. The changing face of the family Enterobacteriaceae (order: "Enterobacterales"): new members, taxonomic issues, geographic expansion, and new diseases and disease syndromes. Clin Microbiol Rev. 2021; 34(2):e00174-00120

Kubota H, Nakayama T, Ariyoshi T, Uehara S, Uchitani Y, Tsuchida S, et al. Emergence of *Phytobacter diazotrophicus* carrying an IncA/C2 plasmid harboring bla_{NDM-1} in Tokyo, Japan. mSphere. 2023; 8(4), e0014723.

Liu L, Feng Y, Wei L, Qiao F, Zong Z. Precise species identification and taxonomy update for the genus *Kluyvera* with reporting *Kluyvera sichuanensis* sp. nov. Front Microbiol. 2020; 11 (579306): 1-10.

Manual of Clinical Microbiology. 13 Ed. Washington, DC: ASM Press: 2023.

Munson E, Carroll KC. Summary of Novel Bacterial Isolates Derived from Human Clinical Specimens and Nomenclature Revisions Published in 2018 and 2019. J Clin Microbiol. 2021; 59(2):e01309-20.

Smits TH, Arend LN, Cardew S, Tång-Hallbäck E, Mira MT, Moore ER, et al. Resolving taxonomic confusion: establishing the genus *Phytobacter* on the list of clinically relevant Enterobacteriaceae. Eur J Clin Microbiol Infect Dis. 2022;41(4):547-58.

Pillonetto M, Arend L, Gomes SM, Oliveira MA, Timm LN, Martins AF, et al. Molecular investigation of isolates from a multistate polymicrobial outbreak associated with contaminated total parenteral nutrition in Brazil. BMC Infect Dis. 2018a;18:397.

Pillonetto M, Arend L, Faoro H, D'Espindula HR, Blom J, Smits TH, et al. Emended description of the genus *Phytobacter*, its type species *Phytobacter diazotrophicus* (Zhang 2008) and description of *Phytobacter ursingii* sp. nov. Int J Syst Evol Microbiol. 2018b; 68(1):176-84.

Building of a New Spectra® for the Identification of *Phytobacter* spp, an Emerging Enterobacterales, Using Maldi Biotyper®

Dal Lin A^{1,2} ; Kulek DO^{1,2}; Gonçalves GA¹; Kraft L¹, Neto JFC¹; Vizontainer G²; Pillonetto M.^{1,2}

¹Laboratório Central do Estado do Paraná, LACEN/PR. São José dos Pinhais, Parana, Brazil

²Pontificia Universidade Católica do Paraná. Curitiba, Paraná, Brazil

Phytobacter spp is an emerging pathogen responsible for different human infections (i.e., sepsis and UTI), including outbreaks (Smits T et al., 2022). The first reported human infection was an outbreak of contaminated Total Parenteral Nutrition in Brazil in 2013 (Pillonetto M et al., 2018a). Recently, it was recognized as an emerging pathogen by Janda & Abbott (2021), Munson & Carroll (2021), and one of the most expressive textbooks of Microbiology - the Manual of Clinical Microbiology (2023). Most identification methods, such as manual biochemical tests, different automation panels, and MALDI-TOF, fail to identify this genus, naming these isolates equivocally as *Pantoea agglomerans*, *Kluyvera intermedia*, *Leclercia adecarboxylata*, *Salmonella* sp., *Klebsiella aerogenes*, *Klebsiella oxytoca*, *Pluralibacter gergoviae*, *Cronobacter sakazakii*, *Cronobacter turicensis* and *Cronobacter* sp. (Pillonetto M et al., 2018a; Pillonetto M et al. 2018b, Smits T et al., 2022; Hon P et al. 2023; Kubota H et al., 2023). Even sequencing methods can misidentify it as *Pseudomonas aeruginosa* and “*Citrobacter bitterii*” (16S rDNA) or *K. intermedia* and *Enterobacter* spp. (Whole Genome Sequencing – WGS) (Liu L et al., 2020; Smits T et al., 2022; Pillonetto M, personal communication). Therefore, there is an urgent need to optimize these identification systems. The present study aimed to build a new MALDI library to identify *Phytobacter* spp correctly.

We used MALDI Biotyper® Sirius (Bruker Daltonics, Bremen, Germany) to construct a new Spectra®. Strains of *P. ursingii*, *P. diazotrophicus*, and *Phytobacter* sp. nov. (not yet published) were used. The isolates came from a reference laboratory in Brazil (LACEN/PR), and all strains had their identification confirmed previously by WGS. The mass spectra were analyzed and compared with other strains in the reference database. Tests were carried out in octuplicate using 19 strains among the three species. The indirect method was used to obtain better quality spectra, where an extraction with acetonitrile was performed before submitting the strains to MALDI, according to the methodology recommended by the fabricants. The software used for data analysis was Bruker Compass® DataAnalysis 2.0, version 3.5.

Differences between species were observed. Figure 1 shows that *P. diazotrophicus* has spectral peaks from 3,000 to 11,000. For *P. ursingii*, the peaks are more stable and linear, starting at 2,000 to 11,000. Furthermore, for the *Phytobacter* sp. nov, mass formation starts at 3,000 and goes up to 16,000, giving a unique profile.

According to our knowledge, Vitek-MS® MALDI-TOF (bioMerieux, France) is still missing the public IVD database for *Phytobacter* spp identification. The exception is an RUO database created successfully at our lab (LACEN/PR), where it correctly identified 96% (24/25) of the isolates (Mazzetti A et al. 2024 – manuscript in preparation). Conversely, a mass profile of *Phytobacter ursingii* was recently uploaded to a new database for MALDI Biotyper® Sirius. This actualization occurred in April 2023 and already supported the identification of two outbreaks caused by *Phytobacter diazotrophicus* in a Kidney Diseases Clinic in Brazil (Gonçalves GA et al., 2023 – submitted) and also in a NICU in Japan (Kubota et al., 2023), as well as a sepsis case in Iowa, USA (Choice S et al. 2024), that still

need to be confirmed, regarding its real species, once WGS was not processed. Although the current profile cannot differentiate between species, it already alerts to the suspicion of the genera *Phytobacter*, guaranteeing that further molecular tests (16S rDNA or WGS) are needed to confirm its identification at the species level. In Japan's outbreak, it was interesting to observe that a first run in MALDI Biotyper® Sirius gave as possible results: *Cronobacter sakazakii* or *Pluralibacter gergoviae* with low identification scores (<2.0). But, on a second run, after the equipment database was updated with the MBT Compass reference library, MBT-BDAL-10833 (Bruker), the identification was *Phytobacter ursingii* with a score ≥ 1.99 for all four isolates. WGS confirmed it was a *Phytobacter* spp, but belonging to a different species - *P. diazotrophicus*. We expect that with the introduction of the present proposed mass profile, it will be possible to differentiate *Phytobacter* species, especially the yet-to-be-validatedly published "*Phytobacter brasiliensis* sp. nov." because it has very distinctive peaks at m/z 1544 and 1606.

Correct identification of *Phytobacter* spp, increasingly reported worldwide, is paramount. Its misidentification with other Enterobacterales can mask outbreak investigation and even induce incorrect treatments or source investigation. Although many strains still need to be tested for an exact differentiation between species, a close analysis of the individual characteristics of the mass weights between the three species is possible. This is an essential advance for including the species *P. diazotrophicus* and "*Phytobacter brasiliensis* sp. nov." in the Biotyper® database since there is only a library for *P. ursingii*, which has also been improved in this study.

Figure 1 - Multiple Spectra Report of three different *Phytobacter* species, showing clear different spectra between them.

References

Choice S, et al. BMJ Case Rep 2024;17:e258384. doi:10.1136/bcr-2023-258384

Hon P, Ko KKK, Zhong JCW, De PP, Smits THM, Low J, Vasoo S, Tsui CKM. Genomic Identification of Two *Phytobacter diazotrophicus* Isolates from a Neonatal Intensive Care Unit in Singapore. Microbiol Resour Announc. 2023 Jun 20;12(6):e0016723. doi: 10.1128/mra.00167-23.

Janda JM, Abbott SL. The changing face of the family Enterobacteriaceae (order: "Enterobacterales"): new members, taxonomic issues, geographic expansion, and new diseases and disease syndromes. Clin Microbiol Rev. 2021; 34(2):e00174-00120

Kubota H, Nakayama T, Ariyoshi T, Uehara S, Uchitani Y, Tsuchida S, et al. Emergence of *Phytobacter diazotrophicus* carrying an IncA/C2 plasmid harboring blaNDM-1 in Tokyo, Japan. mSphere. 2023; 8(4), e0014723.

Liu L, Feng Y, Wei L, Qiao F, Zong Z. Precise species identification and taxonomy update for the genus *Kluyvera* with reporting *Kluyvera sichuanensis* sp. nov. Front Microbiol. 2020; 11 (579306): 1-10.

Manual of Clinical Microbiology. 13 Ed. Washington, DC: ASM Press: 2023.

Munson E, Carroll KC. Summary of Novel Bacterial Isolates Derived from Human Clinical Specimens and Nomenclature Revisions Published in 2018 and 2019. J Clin Microbiol. 2021; 59(2):e01309-20.

Smits TH, Arend LN, Cardew S, Tång-Hallbäck E, Mira MT, Moore ER, et al. Resolving taxonomic confusion: establishing the genus *Phytobacter* on the list of clinically relevant Enterobacteriaceae. Eur J Clin Microbiol Infect Dis. 2022;41(4):547-58.

Pillonetto M, Arend L, Gomes SM, Oliveira MA, Timm LN, Martins AF, et al. Molecular investigation of isolates from a multistate polymicrobial outbreak associated with contaminated total parenteral nutrition in Brazil. BMC Infect Dis. 2018a;18:397.

Pillonetto M, Arend L, Faoro H, D'Espindula HR, Blom J, Smits TH, et al. Emended description of the genus *Phytobacter*, its type species *Phytobacter diazotrophicus* (Zhang 2008) and description of *Phytobacter ursingii* sp. nov. Int J Syst Evol Microbiol. 2018b; 68(1):176-84.

Reviewer 1 – Review Attachment 1

- 1. How many isolates were being used to establish the MALDI-tof MS library? Were type species/isolates included? From different countries? How did you validate the ID initially before using WGS? e.g any phenotypic features on agar compared to other Gram -ve**

19 isolates from three species were used, all of them from Brazil. *Phytobacter diazotrophicus* ATCC 27981, ATCC 27990 and DSM17806 (Type strain) were included. Initial suspected strains were gram-negative, strong lactose fermenting, with triple negative results for lysine, ornithine and arginine, positive for indol, citrate and VP, but negative for the fermentation of inositol and melibiose. All this information has been included in the text.

- 2. Better to describe an example on how the *Phytobacter* isolates were misidentified using 16S and WGS data.**

Since data on GenBank are not curated and this species was only validated in 2017, there are many 16S and WGS sequences deposited previously with the wrong ID, such as many *Kluyvera intermedia* WGS that has been assigned only recently to *Phytobacter ursingii* (i.ex.: CAV1151)

We change the text to be more assertive.

- 3. The fact that a library is available for *P. ursingii* should be moved to the introduction (now it was mentioned in the last sentence). Have you compared the spectra of your *P. ursingii* strains to the commercial available spectra?**

Suggestion accepted. All *Phytobacter ursingii* from our collection were correctly identified by Maldi Biotyper after new library creation. But the existing commercial spectra failed to identify some strains (see new Supplemental Table 1A and 1B).

- 4. MALDI, MALDI tof , MALDI tof MS were used interchangeably. I believe they meant different things, and sometimes they were used incorrectly. The technique should be called MALDI-tof MS.**

Nomenclature changed to the correct: MALDI-TOF-MS.

- 5. What is the x- and y- axes of the MS spectra (Fig.1)? The report of spectra variation (3rd paragraph) should be revised properly. These are peptides of m/z showing different intensity.**

The x -axes is the m/z and the y-axes is the intensity of the peak.

6. is it possible to include the scores of these validation strains in the supplementary materials?

Thanks for this relevant suggestion. We created a supplemental table with all the results from the previous scores and using the new library scores in comparison.

7. Three references were not yet published. They should be removed or cited as (pers comm), and discussed with caution.

Changes made according to the suggestion.

Reviewer 2 – Review Attachment 2

“manual biochemical tests, different automation panels, and MALDI-TOF”

- 1) give examples (e.g. API20/50, Phoenix100, Bruker/Saramis)

Examples added to the text: r/b tubes, API20E, Vitek GN panels, Microscan Plates, and bioMerieux MALDI-TOF-MS

“naming these isolates equivocally as”

- 2) assigning these isolates mistakingly to

Text modified accordingly

“Even sequencing methods”...

- 2) when not properly implemented

Text modified accordingly.

“19 strains among the three species.”

- 3) distributed in which way? please give exact numbers for each species.

03 strains of *Phytobacter sp. nov.*, 04 *P. ursingii* and 12 *P. diazotrophicus*.
Text included accordingly.

“Although the current profile cannot differentiate between species”...

- 4) you mean here the *P. ursingii* profile that is currently with the Bruker system? I'd reverse the sentence to highlight the added value of your new spectra. "Although the current profile can alert to the suspicion of *Phytobacter*, it cannot differentiate between species."

Thanks for the relevant suggestion. Text modified accordingly.

“1544 and 1606”.

- 5) these peaks cannot be seen in Fig. 1. Please extend the lower mass range in the figure. Are all others not distinctive?

In fact the numbers were wrong. The correct peaks number are 15441 and 16066.

“which has also been improved in this study”...

7) How?

“Figure 1 - Multiple Spectra Report of three different *Phytobacter* species, showing clear different spectra between them.”

- 8) Are the highlighted peaks exactly those relevant for the Bruker system or there s more (e.g. 1544 and 1606)?

This are the more relevant peaks founded in this study for the Bruker System.

Reviewer 2 – Review Attachment 1

“Even sequencing methods can misidentify it as *Pseudescherichia vulneris* and “*Citrobacter bitterni*” (16S rDNA) or *K. intermedia* and *Enterobacter* spp. (Whole Genome Sequencing – WGS)”

- 1) Elaborate how it was misidentified? e.g. 16S rDNA blast search , MLST db does not have *Phytobacter*

“Even sequencing methods can misidentify it as *Pseudescherichia vulneris* and “*Citrobacter bitterni*” (16S rDNA), using BLAST or leBIBI searches, or *K. intermedia* and *Enterobacter* spp. (Whole Genome Sequencing – WGS BLAST)”

Text modified accordingly

“*Phytobacter* sp. Nov.”

- 2) *Novel, undescribed Phytobacter species?*

Exactly. A manuscript is being prepared to submit the description of this new species to IJSEM, soon,

“The isolates came from a reference laboratory in Brazil (LACEN/PR)”

- 3) *Have you included the type isolate?*

Phytobacter diazotrophicus ATCC 27981, ATCC 27990 and DSM17806 (Type strain) were included (see text and supplemental Table)

“The indirect method was used to obtain better quality spectra”...

- 4) Any reference?

Text included: as manufacture's recomendation

“That still need to be confirmed, regarding its real species, once WGS was not processed”

5) This sentence was confusing!

Thank you for the observation. The sentence has been corrected.

“In Japan's outbreak”

6) Reference?

Kubota et al. Reference included.

Building of a New Spectra[®] for the Identification of *Phytobacter* spp, an Emerging Enterobacterales, Using Maldi Biotyper[®]

Dal Lin A^{1,2} ; Kulek DO^{1,2}; Gonçalves GA¹; Kraft L¹, Neto JFC¹; Vizentainer G²; Pilonetto M.^{1,2}

¹Laboratório Central do Estado do Paraná, LACEN/PR. São José dos Pinhais, Parana, Brazil

²Pontificia Universidade Católica do Paraná. Curitiba, Paraná, Brazil

Phytobacter spp is an emerging pathogen responsible for different human infections (i.e., sepsis and UTI), including outbreaks (Smits T et al., 2022). The first reported human infection was an outbreak of contaminated Total Parenteral Nutrition in Brazil in 2013 (Pilonetto M et al., 2018a). Recently, it was recognized as an emerging pathogen by Janda & Abbott (2021), Munson & Carroll (2021), and one of the most expressive textbooks of Microbiology - the Manual of Clinical Microbiology (2023). Most identification methods, such as manual biochemical tests, different automation panels, and MALDI-TOF, fail to identify this genus, naming these isolates equivocally as *Pantoea agglomerans*, *Kluyvera intermedia*, *Leclercia adecarboxylata*, *Salmonella* sp., *Klebsiella aerogenes*, *Klebsiella oxytoca*, *Pluralibacter gergoviae*, *Cronobacter sakazakii*, *Cronobacter turicensis* and *Cronobacter* sp. (Pilonetto M et al., 2018a; Pilonetto M et al. 2018b, Smits T et al., 2022; Hon P et al. 2023; Kubota H et al., 2023). Even sequencing methods can misidentify it as *Pseudomonas* spp. or *Citrobacter* spp. (Whole Genome Sequencing – WGS) (Liu L et al., 2020; Smits T et al., 2022; Pilonetto M, personal communication). Therefore, there is an urgent need to optimize these identification systems. The present study aimed to build a new MALDI library to identify *Phytobacter* spp correctly.

We used MALDI Biotyper[®] Sirius (Bruker Daltonics, Bremen, Germany) to construct a new Spectra[®]. Strains of *P. ursingii*, *P. diazotrophicus*, and *Phytobacter* sp. nov. (not yet published) were used. The isolates came from a reference laboratory in Brazil (LACEN/PR), and all strains had their identification confirmed previously by WGS. The mass spectra were analyzed and compared with other strains in the reference database. Tests were carried out in octuplicate using 10 strains among the three species. The indirect method was used to obtain better quality spectra, where an extraction with acetonitrile was performed before submitting the strains to MALDI, according to the methodology recommended by the fabricants. The software used for data analysis was Bruker Compass[®] DataAnalysis 2.0, version 3.5.

Differences between species were observed. Figure 1 shows that *P. diazotrophicus* has spectral peaks from 3,000 to 11,000. For *P. ursingii*, the peaks are more stable and linear, starting at 2,000 to 11,000. Furthermore, for the *Phytobacter* sp. nov, mass formation starts at 3,000 and goes up to 16,000, giving a unique profile.

According to our knowledge, Vitek-MS[®] MALDI-TOF (bioMérieux, France) is still missing the public IVD database for *Phytobacter* spp identification. The exception is the RUO database created successfully at our lab (LACEN/PR), where it correctly identified 96% (24/25) of the isolates (Mazzetti A et al. 2024 – manuscript in preparation). Conversely, a mass profile of *Phytobacter ursingii* was recently uploaded to a new database for MALDI Biotyper[®] Sirius. This actualization occurred in April 2023 and already supported the identification of two outbreaks caused by *Phytobacter diazotrophicus* in a Kidney Diseases Clinic in Brazil (Gonçalves GA et al., 2023 – submitted) and also in a NICU in Japan (Kubota et al., 2023), as well as a sepsis case in Iowa, USA (Choice S et al. 2024), that still

need to be confirmed, regarding its real species, because WGS was not processed. Although the current profile cannot differentiate between species, it already alerts to the suspicion of the genera *Phytobacter*, guaranteeing that further molecular tests (16S rDNA or WGS) are needed to confirm its identification at the species level. In Japan's outbreak, it was interesting to observe that a first run in MALDI Biotyper® Sirius gave as possible results: *Cronobacter sakazakii* or *Pluralibacter gergoviae* with low identification scores (<2.0). But, on a second run, after the equipment database was updated with the MBT Compass reference library, MBT-BDAL-10833 (Bruker), the identification was *Phytobacter ursingii* with a score ≥ 1.99 for all four isolates. WGS confirmed it was a *Phytobacter* spp, but belonging to a different species - *P. diazotrophicus*. We expect that with the introduction of the present proposed mass profile, it will be possible to differentiate *Phytobacter* species, especially the yet-to-be-validatedly published "*Phytobacter brasiliensis* sp. nov." because it has very distinctive peaks at m/z 1544 and 1606.

Correct identification of *Phytobacter* spp, increasingly reported worldwide, is paramount. Its misidentification with other Enterobacterales can mask outbreak investigation and even induce incorrect treatments or source investigation. Although many strains still need to be tested for an exact differentiation between species, a close analysis of the individual characteristics of the mass weights between the three species is possible. This is an essential advance for including the species *P. diazotrophicus* and "*Phytobacter brasiliensis* sp. nov." in the Biotyper® database since there is only a library for *P. ursingii*, which has also been improved in this study.

Figure 1 - Multiple Spectra Report of three different *Phytobacter* species, showing clear different spectra between them.

References

Choice S, et al. BMJ Case Rep 2024;17:e258384. doi:10.1136/bcr-2023-258384

Hon P, Ko KKK, Zhong JCW, De PP, Smits THM, Low J, Vasoo S, Tsui CKM. Genomic Identification of Two *Phytobacter diazotrophicus* Isolates from a Neonatal Intensive Care Unit in Singapore. Microbiol Resour Announc. 2023 Jun 20;12(6):e0016723. doi: 10.1128/mra.00167-23.

Janda JM, Abbott SL. The changing face of the family Enterobacteriaceae (order: "Enterobacterales"): new members, taxonomic issues, geographic expansion, and new diseases and disease syndromes. Clin Microbiol Rev. 2021; 34(2):e00174-00120

Kubota H, Nakayama T, Ariyoshi T, Uehara S, Uchitani Y, Tsuchida S, et al. Emergence of *Phytobacter diazotrophicus* carrying an IncA/C2 plasmid harboring bla_{NDM-1} in Tokyo, Japan. mSphere. 2023; 8(4), e0014723.

Liu L, Feng Y, Wei L, Qiao F, Zong Z. Precise species identification and taxonomy update for the genus *Kluyvera* with reporting *Kluyvera sichuanensis* sp. nov. Front Microbiol. 2020; 11 (579306): 1-10.

Manual of Clinical Microbiology. 13 Ed. Washington, DC: ASM Press: 2023.

Munson E, Carroll KC. Summary of Novel Bacterial Isolates Derived from Human Clinical Specimens and Nomenclature Revisions Published in 2018 and 2019. J Clin Microbiol. 2021; 59(2):e01309-20.

Smits TH, Arend LN, Cardew S, Tång-Hallbäck E, Mira MT, Moore ER, et al. Resolving taxonomic confusion: establishing the genus *Phytobacter* on the list of clinically relevant Enterobacteriaceae. Eur J Clin Microbiol Infect Dis. 2022;41(4):547-58.

Pillonetto M, Arend L, Gomes SM, Oliveira MA, Timm LN, Martins AF, et al. Molecular investigation of isolates from a multistate polymicrobial outbreak associated with contaminated total parenteral nutrition in Brazil. BMC Infect Dis. 2018a;18:397.

Pillonetto M, Arend L, Faoro H, D'Espindula HR, Blom J, Smits TH, et al. Emended description of the genus *Phytobacter*, its type species *Phytobacter diazotrophicus* (Zhang 2008) and description of *Phytobacter ursingii* sp. nov. Int J Syst Evol Microbiol. 2018b; 68(1):176-84.

Building of a New Spectra® for the Identification of *Phytobacter* spp, an Emerging Enterobacterales, Using Maldi Biotyper®

Dal Lin A^{1,2} ; Kulek DO^{1,2}; Gonçalves GA¹; Kraft L¹, Neto JFC¹; Vizontainer G²; Pillonetto M.^{1,2}

¹Laboratório Central do Estado do Paraná, LACEN/PR. São José dos Pinhais, Parana, Brazil

²Pontificia Universidade Católica do Paraná. Curitiba, Paraná, Brazil

Phytobacter is an emerging pathogen responsible for different human infections (i.e., sepsis and UTI), including outbreaks (Smits T et al., 2022). The first reported human infection was an outbreak of contaminated Total Parenteral Nutrition in Brazil in 2013 (Pillonetto M et al., 2018a). Recently, it was recognized as an emerging pathogen by Janda & Abbott (2021), Munson & Carroll (2021), and one of the most expressive textbooks of Microbiology - the Manual of Clinical Microbiology (2023). Most identification methods, such as manual biochemical tests, different automation panels, and MALDI-TOF, fail to identify this genus, naming these isolates equivocally as *Pantoea agglomerans*, *Kluyvera intermedia*, *Leclercia adecarboxylata*, *Salmonella* sp., *Klebsiella aerogenes*, *Klebsiella oxytoca*, *Pluralibacter gergoviae*, *Cronobacter sakazakii*, *Cronobacter turicensis* and *Cronobacter* sp. (Pillonetto M et al., 2018a; Pillonetto M et al. 2018b, Smits T et al., 2022; Van P et al. 2023; Kubota H et al., 2023). Even sequencing methods can misidentify it as *Pseudomonas aeruginosa* and “*Citrobacter bitterii*” (16S rDNA) or *K. intermedia* and *Enterobacter* spp. (Whole Genome Sequencing – WGS) (Liu L et al., 2020; Smits T et al., 2022; Pillonetto M, personal communication). Therefore, there is an urgent need to optimize these identification systems. The present study aimed to build a new MALDI library to identify *Phytobacter* spp correctly.

We used MALDI Biotyper® Sirius (Bruker Daltonics, Bremen, Germany) to construct a new Spectra®. Strains of *P. ursingii*, *P. diazotrophicus*, and *Phytobacter* sp. nov. (not yet published) were used. The isolates came from a reference laboratory in Brazil (LACEN/PR), and all strains had their identification confirmed previously by WGS. The mass spectra were analyzed and compared with other strains in the reference database. Tests were carried out in octuplicate using 19 strains among the three species. The direct method was used to obtain better quality spectra, where an extraction with acetonitrile was performed before submitting the strains to MALDI, according to the methodology recommended by the fabricants. The software used for data analysis was Bruker Compass® DataAnalysis 2.0, version 3.5.

Differences between species were observed. Figure 1 shows that *P. diazotrophicus* has spectral peaks from 3,000 to 11,000. For *P. ursingii*, the peaks are more stable and linear, starting at 2,000 to 11,000. Furthermore, for the *Phytobacter* sp. nov, mass formation starts at 3,000 and goes up to 16,000, giving a unique profile.

According to our knowledge, Vitek-MS® MALDI-TOF (bioMerieux, France) is still missing the public IVD database for *Phytobacter* spp identification. The exception is an RUO database created successfully at our lab (LACEN/PR), where it correctly identified 96% (24/25) of the isolates (Mazzetti A et al. 2024 – manuscript in preparation). Conversely, a mass profile of *Phytobacter ursingii* was recently uploaded to a new database for MALDI Biotyper® Sirius. This actualization occurred in April 2023 and already supported the identification of two outbreaks caused by *Phytobacter diazotrophicus* in a Kidney Diseases Clinic in Brazil (Gonçalves GA et al., 2023 – submitted) and also in a NICU in Japan (Kubota et al., 2023), as well as a sepsis case in Iowa, USA (Choice S et al. 2024), that still

id to be confirmed, regarding its real species, once WGS was not processed. Although the current profile cannot differentiate between species, it already alerts to the suspicion of the genera *Phytobacter*, guaranteeing that further molecular tests (16S rDNA or WGS) are needed to confirm its identification at the species level. In Japan's outbreak, it was interesting to observe that a first run in MALDI Biotyper® Sirius gave as possible results: *Cronobacter sakazakii* or *Pluralibacter gergoviae* with low identification scores (<2.0). But, on a second run, after the equipment database was updated with the MBT Compass reference library, MBT-BDAL-10833 (Bruker), the identification was *Phytobacter ursingii* with a score ≥ 1.99 for all four isolates. WGS confirmed it was a *Phytobacter* spp, but belonging to a different species - *P. diazotrophicus*. We expect that with the introduction of the present proposed mass profile, it will be possible to differentiate *Phytobacter* species, especially the yet-to-be-validatedly published "*Phytobacter brasiliensis* sp. nov." because it has very distinctive peaks at m/z 1544 and 1606.

Correct identification of *Phytobacter* spp, increasingly reported worldwide, is paramount. Its misidentification with other Enterobacterales can mask outbreak investigation and even induce incorrect treatments or source investigation. Although many strains still need to be tested for an exact differentiation between species, a close analysis of the individual characteristics of the mass weights between the three species is possible. This is an essential advance for including the species *P. diazotrophicus* and "*Phytobacter brasiliensis* sp. nov." in the Biotyper® database since there is only a library for *P. ursingii*, which has also been improved in this study.

Figure 1 - Multiple Spectra Report of three different *Phytobacter* species, showing clear different spectra between them.

References

Choice S, et al. BMJ Case Rep 2024;17:e258384. doi:10.1136/bcr-2023-258384

Hon P, Ko KKK, Zhong JCW, De PP, Smits THM, Low J, Vasoo S, Tsui CKM. Genomic Identification of Two *Phytobacter diazotrophicus* Isolates from a Neonatal Intensive Care Unit in Singapore. Microbiol Resour Announc. 2023 Jun 20;12(6):e0016723. doi: 10.1128/mra.00167-23.

Janda JM, Abbott SL. The changing face of the family Enterobacteriaceae (order: "Enterobacterales"): new members, taxonomic issues, geographic expansion, and new diseases and disease syndromes. Clin Microbiol Rev. 2021; 34(2):e00174-00120

Kubota H, Nakayama T, Ariyoshi T, Uehara S, Uchitani Y, Tsuchida S, et al. Emergence of *Phytobacter diazotrophicus* carrying an IncA/C2 plasmid harboring bla_{NDM-1} in Tokyo, Japan. mSphere. 2023; 8(4), e0014723.

Liu L, Feng Y, Wei L, Qiao F, Zong Z. Precise species identification and taxonomy update for the genus *Kluyvera* with reporting *Kluyvera sichuanensis* sp. nov. Front Microbiol. 2020; 11 (579306): 1-10.

Manual of Clinical Microbiology. 13 Ed. Washington, DC: ASM Press: 2023.

Munson E, Carroll KC. Summary of Novel Bacterial Isolates Derived from Human Clinical Specimens and Nomenclature Revisions Published in 2018 and 2019. J Clin Microbiol. 2021; 59(2):e01309-20.

Smits TH, Arend LN, Cardew S, Tång-Hallbäck E, Mira MT, Moore ER, et al. Resolving taxonomic confusion: establishing the genus *Phytobacter* on the list of clinically relevant Enterobacteriaceae. Eur J Clin Microbiol Infect Dis. 2022;41(4):547-58.

Pillonetto M, Arend L, Gomes SM, Oliveira MA, Timm LN, Martins AF, et al. Molecular investigation of isolates from a multistate polymicrobial outbreak associated with contaminated total parenteral nutrition in Brazil. BMC Infect Dis. 2018a;18:397.

Pillonetto M, Arend L, Faoro H, D'Espindula HR, Blom J, Smits TH, et al. Emended description of the genus *Phytobacter*, its type species *Phytobacter diazotrophicus* (Zhang 2008) and description of *Phytobacter ursingii* sp. nov. Int J Syst Evol Microbiol. 2018b; 68(1):176-84.

Re: Spectrum01107-24R1 (Building of a New Spectra® for the Identification of *Phytobacter* spp, an Emerging Enterobacterales, Using Maldit Biotyper®)

Dear Dr. Marcelo Pillonetto:

Thank you for the privilege of reviewing your work. Below you will find my comments, instructions from the Spectrum editorial office, and the reviewer comments.

Thank you for responding to the Reviewers' comments. I think the addition of the supplemental tables adds significantly to the manuscript. Before the paper can move forward for publication, the following minor issues require attention. Once these are addressed, I will be happy to move the paper forward for publication. A tracked changed version of the manuscript is included for convenience.

- 1) There are numerous spelling and grammatical errors. See attached document.
- 2) Please include the manufacturer name and location for all commercial systems listed.
- 3) Line 17, please define r/b tubes and note the manufacturer.
- 4) Line 30-31, please add the ATCC strains and DMZ type strain to the text.
- 5) p. 2, lines 6-7, this is still confusing as written. Please clarify the highlighted phrase.
- 6) With respect to the references, please note the specific chapter in MCM for reference 5 and likewise for reference 10, please include the title of the article.

Revision Guidelines

Sincerely,
Karen Carroll

Response to the Reviewers - Spectrum01107-24R1 Decision Letter

Dear Editor/Reviewers,

Thank you kindly for all your contributions

- 1) There are numerous spelling and grammatical errors. See attached document.

All grammatical errors have been corrected as shown in the attached marked document. Thanks for all the observations.

- 2) Please include the manufacturer name and location for all commercial systems listed.

Manufacturer names and locations have been included accordingly.

- 3) Line 17, please define r/b tubes and note the manufacturer.

We changed the r/b tubes to its actual commercial name and included the manufacturer as well.

- 4) Line 30-31, please add the ATCC strains and DMZ type strain to the text.

ATCC and DSM type strains added to the text (line 35 now).

- 5) p. 2, lines 6-7, this is still confusing as written. Please clarify the highlighted phrase.

We rewrote the phrase to better clarity. Please see if it is more comprehensive now. The authors (Choice S et al.) only did MALDI-TOF for identification. I have contacted them by e-mail, and they confirmed this information. Also, they don't have the strain anymore, unfortunately.

- 6) With respect to the references, please note the specific chapter in MCM for reference 5 and likewise for reference 10, please include the title of the article.

Thanks for noticing. We added the authors and chapter's name accordingly.

Marcelo Pilonetto
Curitiba, July 29, 2024.

Re: Spectrum01107-24R2 (Building of a New Spectra® for the Identification of *Phytobacter* spp, an Emerging Enterobacterales, Using Maldi Biotyper®)

Dear Dr. Marcelo Pillonetto:

Thank you for making the additional minor edits. I think they improved the clarity of the paper. The manuscript is now ready for publication.

Your manuscript has been accepted, and I am forwarding it to the ASM production staff for publication. Your paper will first be checked to make sure all elements meet the technical requirements. ASM staff will contact you if anything needs to be revised before copyediting and production can begin. Otherwise, you will be notified when your proofs are ready to be viewed.

Sincerely,
Karen Carroll
Editor
Microbiology Spectrum